# Nondestructive Interface Morphology Characterization of Thermal Barrier Coatings Using Terahertz Time-Domain Spectroscopy

**Dongdong Ye** [1] , **Weize Wang** [1,*], **Jibo Huang** [1], **Xiang Lu** [1] **and Haiting Zhou** [2]

1   Key Lab of Safety Science of Pressurized System, Ministry of Education, School of Mechanical and Power Engineering, East China University of Science and Technology, Shanghai 200237, China; Y10170088@mail.ecust.edu.cn (D.Y.); Y20150084@mail.ecust.edu.cn (J.H.); Y45160090@mail.ecust.edu.cn (X.L.)

2   Department of Quality and Safety Engineering, China Jiliang University, Hangzhou 310018, China; zhouhaiting@cjlu.edu.cn

*   Correspondence: wangwz@ecust.edu.cn; Tel.: +86-21-64252819; Fax: +86-21-64253513

**Abstract:** In this work, a terahertz time-domain spectroscopy (THz-TDS) system was used to measure the thickness of thermal barrier coatings (TBCs) and characterize the interface morphology of TBCs after erosion. Reflection mode, with an angle of incidence of 0, was used for inspection before and after erosion. The refractive index, thickness, and internal structure evolution tendency of the yttria-stabilized zirconia (YSZ) top coat were estimated under consideration of the interaction between the pulsed THz waves and the TBCs. The surface roughness of the top coat surface was considered for the errors analysis in the refractive index and thickness measurement. To reduce the errors introduced by the refractive index change after erosion, two mathematical models were built to assess the thickness loss. Then, the thickness loss was compared with results estimated by the micrometer inspection method. Finally, the basic erosion sample profile with $R_a$ roughness was obtained, and the broadening of THz pulses were suggested as a possible measure for the top coat porosity change, showing that THz waves can be a novel online non-destructive and non-contact evaluation method that can be widely utilized to evaluate the integrity of TBCs applied to gas turbine blades.

**Keywords:** non-destructive; thermal barrier coatings; terahertz time-domain spectroscopy; interface morphology; erosion test

## 1. Introduction

Thermal barrier coatings (TBCs) are widely applied to the metallic surfaces on the gas-turbine or aero-engine parts operating at elevated temperatures, enabling the engines to operate at gas temperatures well above the melting temperature of the superalloy, thereby improving engine efficiency and performance [1]. TBCs typically consist of four layers: The brittle ceramic topcoat (TC), thermally-grown oxide (TGO), metallic bond coat (BC), and metal substrate [1,2]. The TC layer is typically composed of yttria-stabilized zirconia (YSZ), which is expected to remain stable in the operating environments. However, the thinning and peeling-off of TC could be found due to the effect of harsh operating conditions, including erosion, corrosion, and foreign object damage, etc. [1,3,4], which makes the performance of TBCs unsatisfactory. As a result, the wide application of TBCs has been entitled to advanced nondestructive test (NDT) techniques for evaluation and failure detection during service [5–10].

Various NDT techniques have been developed to examine the TBCs up to now, such as ultrasonic waves testing, eddy current testing, rare earth luminescence examination, and X-ray imaging, etc. Some NDT technologies can measure the thickness or monitor the health of TBCs. However, these methods have their own drawbacks. For example, the ultrasound waves are limited to the existence of edge effects and the requirement of a liquid couplant [7,11,12]. The eddy current test signal is susceptible, and the measurement of non-metallic materials was not suitable for the evaluation of complex components as it requires the lift-off operation step, which generates large noise [13–16]. The rare earth luminescence method needs extra doping of rare earth elements and cannot be used to characterize the morphology quantitatively [6,17,18]. X-ray is harmful to the human body due to its high radiant energy [19]. Therefore, it may be inappropriate to use these techniques as the real-time NDT method to characterize the interface morphology of TBCs.

The frequency range of terahertz waves is generally considered to be between 0.1 and 10 THz, which lies between the infrared waves and microwaves. Numerous nonmetallic dielectric material, which is opaque in the range of infrared and visible light, can be penetrated by THz waves. Furthermore, the THz inspection provides a nondestructive, noncontact, nonionizing, and real-time evaluation. It has already been successfully used to characterize materials with composite structures, such as glass fiber reinforced plastic (GFRP) composite [20,21], integrated circuit packages [22], pharmaceutical tablets [23,24], and polymer coated steel [25]. The thickness of TC, the progress of the TGO layer, and stress-induced air-filled voids were also obtained by this method [26–30].

Erosion of TBCs has been considered the secondary primary cause of failure, however, it has become more of an issue and has received more attention [3]. The papers have not been found on the characterization of the coating erosion by THz inspection. In this study, a THz time-domain spectroscopy (THz-TDS) system of the reflection mode was used to measure the thickness of TBCs and characterize the interface morphology of TBCs after erosion.

## 2. Experimental

### 2.1. THz-TDS System and the Inspection Method

In this study, the THz-TDS system (University of Shanghai for Science and Technology, Shanghai, China) was used to characterize the interface morphology of erosion TBCs. The main components of this system included a femtosecond laser module, THz-TDS module, delay line module, emitter module, and receiver module. A schematic of the system is shown in Figure 1. The laser provides pulses of about 80 fs duration at a wavelength of 800 nm. The repeat frequency, and half-height spectrum width and output power of the femtosecond laser generated by the femtosecond laser were 76 MHz, 11 nm, and 1.1 w, respectively. The THz-TDS system was configured in the reflection mode, with an angle of incidence of 0. The spot diameter of terahertz waves was 2 mm. Each data was obtained by 256 scans that were avergaged to ensure reliability. Additionally, to avoid water vapor absorption, dry air (below 1% relative humidity) was supplied to the system. The test temperature was 20 °C. At the beginning of the experiment, the total reflection reference signal without a sample was obtained.

When THz waves were incident on the sample, the surface echo could be detected as shown in Figure 2. Part of the incident terahertz waves was reflected from the topcoat surface, while part of it transmitted through the top coat and was completely reflected at the interface between the top coat and metallic bond coat. Part of the reflected terahertz waves from the interface transmitted through the topcoat surface into air and part of it was reflected back into the topcoat. Multiple reflections took place between the topcoat surface and interface. Here, S is the surface reflection, R1 is the interface reflection, and R2 is the multiple reflections, which made two round trips through the top coat. S embodied the surface message of the top coat and R1 and R2 embodied the inner message of the top coat.

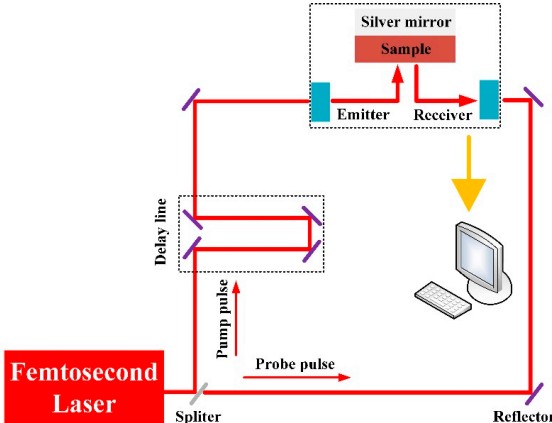

**Figure 1.** Schematic diagram of the THz time-domain spectroscopy (TDS) system in reflection.

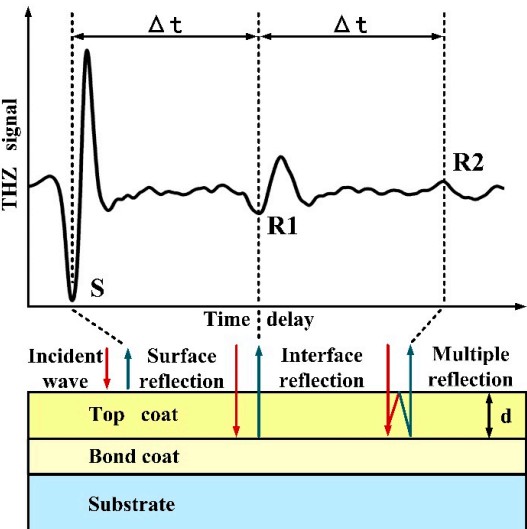

**Figure 2.** Multiple reflections of terahertz waves between the topcoat and bond coat.

The thickness of the top coat can be estimated through the time delay, $\Delta t$, between adjacent reflections and the refractive index of the top coat, $n$, as the following equation:

$$d = \frac{c\Delta t}{2n} \tag{1}$$

Generally, $c$ is the speed of light. For Equation (1), the refractive index, $n$, can be obtained by using the following equation of the relative permittivity and relative permeability of YSZ:

$$n = \sqrt{\varepsilon_r \mu_r} \tag{2}$$

where $\varepsilon_r$ is the relative permittivity of YSZ, which is 27 [31,32], and, generally, $\mu_r$ is the relative permeability of YSZ, which is almost 1 in paramagnetic and diamagnetic materials [22,33]. The refractive index of YSZ obtained by Equation (2) was 5.20. Given the variability of the refractive index with the thermal spray condition, the above refractive index cannot be used directly [34].

The actual refractive index of the top coat was estimated in the frequency region of 0.3 to 0.5 THz by the following Equation (3) [26]:

$$n = 2\frac{F_S F_{R2}}{F_{R1}^2} + 2\sqrt{\left(\frac{F_S F_{R2}}{F_{R1}^2}\right)^2 + \frac{F_S F_{R2}}{F_{R1}^2} + 1} \tag{3}$$

From the reflections shown in Figure 1, the frequency characteristics, $F_S$, $F_{R1}$, and $F_{R2}$, were obtained by taking the fast fourier transform (FFT) for the reflections, S, R1, and R2.

Since the scattering of photons followed longer routes than those that were directly transmitted, the effective optical thickness of the top coat was longer than the geometrical thickness, $d$. The combination of the longer optical path length and dispersion broadened the terahertz pulse. Photons that underwent multiple scattering also accumulated dispersion and therefore influenced the broadening of the pulse. So, the inner structural changes of the top coat may be estimated by the broadening of the pulse.

### 2.2. Preparation of Coatings

In this study, $ZrO_2$ 8 wt.% $Y_2O_3$ (8YSZ) powder (15–45 μm, Chengdu HuaYin Powder Technology Co., Ltd., Chengdu, China) and NiCrAlY powder (45–106 μm, Beijing SunSpraying New Material Co., Ltd., Beijing, China) were used to deposit the top coat and bond coat, respectively. Coatings were applied on a grit-blasted disc-shaped IN738 substrate (Ø 25.4 mm × 3.1 mm). Both the YSZ ceramic coating and bond coat were deposited via a commercial air-plasma spray (APS) system (APS-2000, Beijing Aeronautical Manufacturing Technology Research Institute, Beijing, China). During the whole spraying procedure, argon served as the primary plasma gas, and hydrogen was selected as an auxiliary gas. The pressure of argon and hydrogen were fixed at 0.4 and 0.25 MPa, respectively. Argon was also used as the powder feed gas at a flow rate of 10 L/min. The plasma power was maintained at 36 kW (600 A/60 V) and 30 kW (500 A/60 V) to deposit the YSZ and the bond coat, respectively. The spray gun was operated by the Asea Brown Boveri Ltd. (ABB) manipulator (ABB Group, Zurich, Switzerland) at a speed of 150 mm/s with a spray distance of 70 mm for top coat deposition and of 100 mm for bond coat deposition. For all specimens, the thickness of the top coat was between 150 to 400 μm, while that of the bond coat was about 150 μm.

### 2.3. Erosion Test and the Microstructure Examination

Erosion tests were conducted using the self-made multi-phase erosion platform. The schematic is shown in Figure 3. The working principle of the experimental platform is to generate saturated water vapor with a certain pressure and flow rate by a high-speed steam generator and valves, further carrying the accelerated particles to impact onto the specimen surface through a nozzle. The steam flow rate was about 7 kg/h, and erosion abrasive was 60–75 μm irregular corundum particles. The direction of the steam flow was perpendicular to the coatings' surface, with a powder feeding rate of 5 g/min and a 6 cm nozzle-to-substrate distance was used. To obtain a well-defined geometry erosion scar, a 20 mm diameter opening was used to mask the specimen surface. Each erosion test was controlled for half a minute until the coatings failed. The failure criteria of erosion tests were defined as thinning and peeling by visual inspection.

The as-sprayed coatings and the erosion coatings were examined by THz time-domain spectroscopy. Each coating was tested in three different positions. The thickness and microstructure of coatings were examined by a scanning electron microscope (SEM, ZEISS EVO MA15, Carl Zeiss SMT, Oberkochen, Germany).

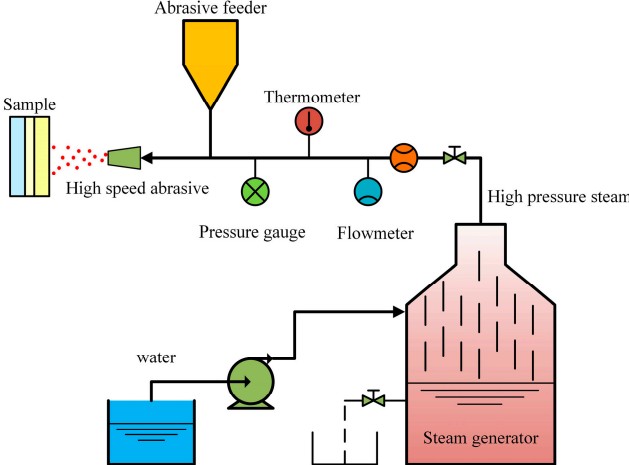

**Figure 3.** Schematic diagram of the self-made multi-phase erosion platform.

## 3. Results and Discussion

### 3.1. Terahertz Properties and Thickness Measurement of Topcoat

Figure 4 shows the THz inspection results of the as-sprayed TBCs specimen and reference without a specimen. S, R1, and R2 in Figure 2 correspond to the first, second, and third reflections in Figure 4, respectively. The time delay between adjacent waveforms ($\Delta t_1$ and $\Delta t_2$) was 13.92 ps and 13.81 ps, respectively. Moreover, several extra THz parameters (e.g., peak to peak value and peak intensity) can also be extracted from Figure 4.

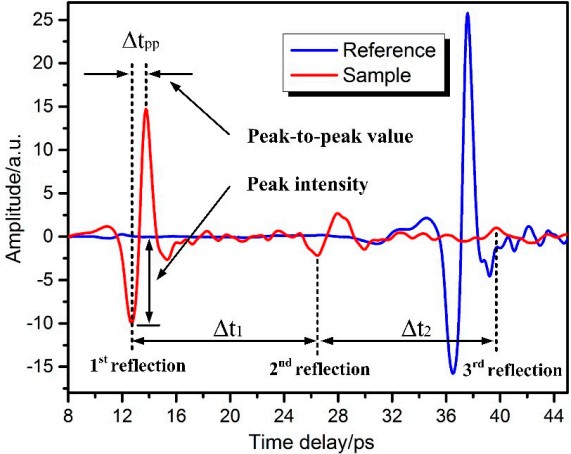

**Figure 4.** Reflected terahertz waveforms obtained from the intact specimen.

The three reflections were extracted from Figure 4, and the frequency characteristics, $F_S$, $F_{R1}$, and $F_{R2}$, obtained by taking the FFT are shown in Figure 5.

According to Equations (1) and (3), the average refractive index of topcoat was 4.98 ± 0.09 at the frequency of 0.32 THz, which corresponded to the maximum of the amplitude sum of $F_S$, $F_{R1}$, and $F_{R2}$, and the average thickness was 418.17 ± 6.26 μm. The thickness obtained from the observation by SEM was 405.68 ± 11.33 μm, as shown in Figure 6. Then, the thickness error of terahertz measurement relative to the SEM observation $(d_{\mathrm{THz}} - d_{\mathrm{SEM}})/d_{\mathrm{SEM}}$ was 3.08%, which proved the reliability of the THz inspection.

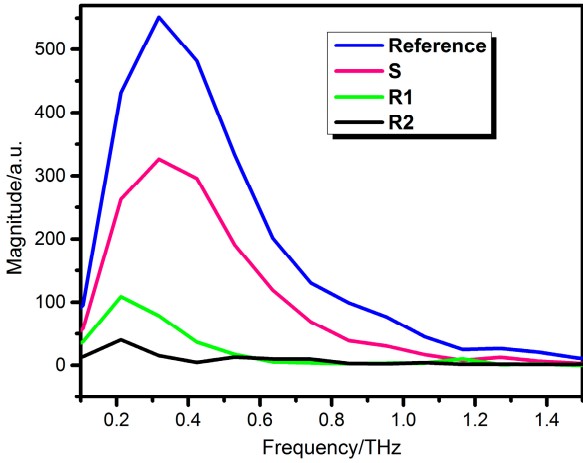

**Figure 5.** Frequency characteristics of reflected waves, S, R1, and R2, obtained from the intact specimen.

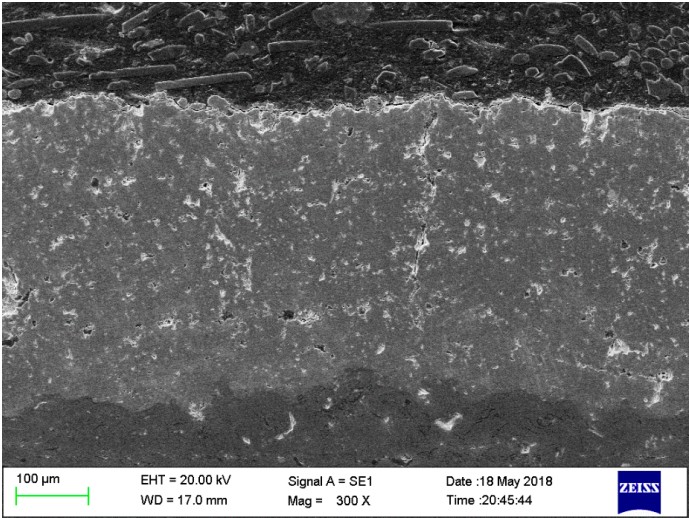

**Figure 6.** Interface microscopic morphology of thermal barrier coatings before erosion.

### 3.2. Error Analysis

When terahertz waves at frequency, $f$, were reflected from the metal surface with $R_z$ roughness, $\sigma$ ($\sigma_{Ra} \approx 0.25\sigma_{Rz}$), the reflectance, $R_r$, and total reflectance, $R_0$, were related by the Kirchhoff Equation, shown as Equation (4), for the errors introduced by surface roughness:

$$\frac{R_r}{R_0} = \mathrm{e}^{-\left(\frac{4\pi\sigma f}{c}\right)^2} \tag{4}$$

According to Figure 5, the optimal frequency value of 0.32 THz, which was used to estimate the refractive index of the top coat, was selected for error analysis. From the Kirchhoff equation, the frequency characteristics of $R_r/R_0$ in the case of $\sigma$ = 5, 7.5, 10, 15, 20, and 30 μm are shown in Figure 7. The value of $R_r/R_0$ was 99.57%, 99.01%, 98.23%, 96.08%, 93.21%, and 85.25% in the case of $\sigma$ = 5, 7.5, 10, 15, 20, and 30 μm when the frequency, $f$, was 0.32 THz.

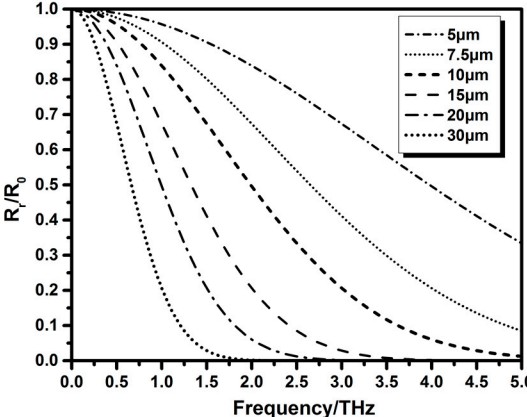

**Figure 7.** Frequency dependence of the effective reflectance, $R_r/R_0$, for different surface roughness.

After considering the influence of surface roughness, Equation (3), used to estimate the refractive index, *n*, was changed to Equation (5):

$$\frac{F_S F_{R2}}{F_{R1}^2} = \frac{R_r}{R_0} \frac{(n-1)^2}{4n} \tag{5}$$

A surface roughness tester (TR200, Beijing Times Peak Technology Co., Ltd., Beijing, China) was used to measure the surface roughness of the top coat. The tester used a diamond stylus of a 10 μm diameter with a precision of 0.01 μm. The average surface, $R_z$, roughness of the top coat was 12.88 μm obtained by the surface roughness tester, and the effective reflectance, $R_r/R_0$, was about 98.04%. The refractive index, when considering the error introduced by the surface roughness, was 5.16 ± 0.10. The corrective average thickness was 403.05 ± 5.88 μm. The relative error $(d_{THz} - d_{SEM})/d_{SEM}$ was only 0.57%, and the measurement accuracy was improved compared with that without considering the effect of roughness.

According to the Fraunhofer standard [35,36], if the surface was judged to be rough, the $R_z$ roughness should be conformed to Equation (6) as follows:

$$\sigma \geq \frac{\lambda}{32 \cos \theta} \tag{6}$$

θ is the incidence angle of the THz waves (θ = 0). According to Equation (6), the critical value of surface roughness was about 29.30 μm at the frequency of 0.32 THz. The Fraunhofer standard and the thickness measurement results above showed that the top coat surface of the as-sprayed sample could be considered as the smooth surface and the effect of surface roughness could be ignored. Despite errors introduced by ignoring the initial roughness of the as-sprayed sample, the terahertz properties and top coat thickness could be easily determined and the errors were in the acceptable range. However, for the eroded coatings, this error should be taken seriously. Nevertheless, the ceramic coatings were deposited on the flat surface here, and the terahertz spots of 1st and 2nd reflection will overlap. But the real turbine blade curved shape cannot keep the incident angle at 0 in most cases, the two spots will not overlap and there will exist a distance *b* (*b* > 0) between them for the real turbine blade curved shape. The distance *b* can be used to estimate the real incident angle θ. The original thickness of ceramic coatings and thickness loss after erosion can be estimated under the consideration of the real incident angle θ for real turbine blades or curved specimens. This problem needs to be solved and the future research will focus on it for the further practical application.

### 3.3. Interface Morphology Characterization

Figure 8 shows the macroscopic morphology of the coat after erosion. The coatings consisted of the un-eroded part and erosion part, in which the initial top coat thickness of the as-sprayed sample obtained by the terahertz method was 177.23 μm before erosion.

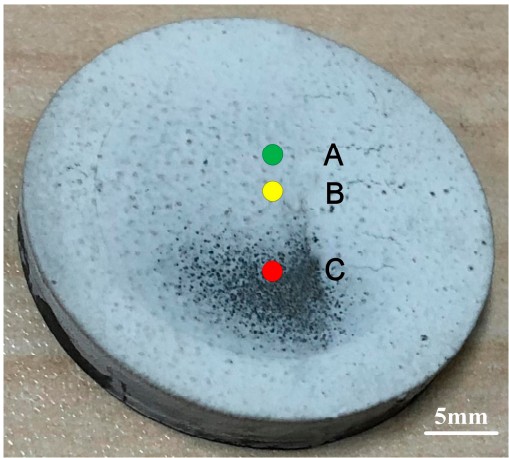

**Figure 8.** Macroscopic morphology of the coat after the erosion test.

Figure 9 shows the terahertz time-domain spectroscopy of the three tested erosion areas in Figure 8 from shallow to deep: A (general erosion area-level 1), B (general erosion area-level 2), and C (over-eroded area). Here, the general erosion area meant that the top coat was remaining in this area and the over eroded area meant that the top coat was completely washed away in this area. Points A, B, and C represent the central spot of the tested areas.

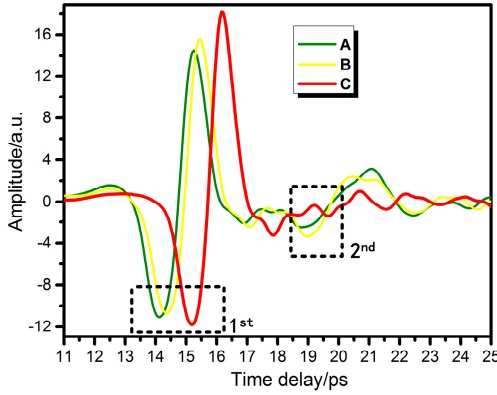

**Figure 9.** Terahertz time-domain spectroscopy with different degrees of erosion.

With the increasing degree of erosion, the positions of the first reflection peak moved back gradually and the thicknesses of the TBCs continued to decrease, which was consistent with reality. Compared with A and B, the amplitude of the first reflection peak of C was greater than A and B, however, the amplitude of the second reflection peak of C was smaller than A and B. This is because the terahertz reflection on the metallic material surface of the bond coat was stronger than the surface of the top coat.

Figure 10 shows the uniform thinning model. The erosion process was assumed as a uniform segmentation thinning and the interface morphology of TBCs was rectangle-shaped.

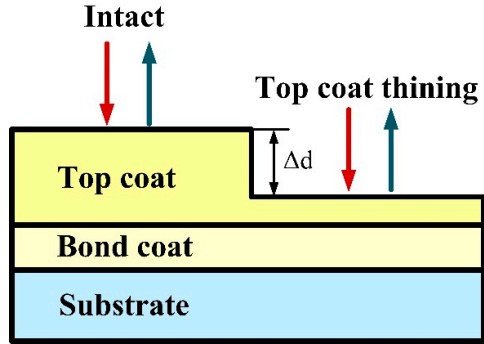

**Figure 10.** Schematic diagram of the uniform thinning model.

The loss of thickness, $\Delta d$, was estimated by Equation (7):

$$\Delta d = \frac{c\Delta T}{2n_{Air}} \tag{7}$$

where $n_{Air}$ is the refractive index of air, which was 1, and $\Delta T$ is the first reflection peak time difference value between erosion areas with intact areas. It should be noted that Equation (1) cannot be used to estimate the loss of thickness, because the refractive index of the top coat had changed as the pores and cracks were introduced by erosion [37]. The loss of thickness can be obtained from the first reflection time, and the time interval is shown in Figure 9, and results are listed in Table 1.

**Table 1.** Time delay, $\Delta T$, and loss of thickness, $\Delta d$, for different erosion positions (A, B, C).

| Experimental Results | Intact | A | B | C |
|---|---|---|---|---|
| First reflection peak time (ps) | $13.71 \pm 0.11$ | 14.13 | 14.35 | 15.20 |
| Time interval $\Delta T$ (ps) | – | 0.43 | 0.64 | 1.49 |
| Loss of thickness $\Delta d$ (μm) | – | 63.99 | 96.00 | 224.00 |
| Electronic digital readout micrometer (μm) | – | 70 | 106 | 228 |

The loss thickness was also examined by an electronic digital readout micrometer and the diameter of the micrometer test needle was also 2 mm, which corresponded to the spot size of the terahertz waves. The measurement comparison with those obtained by terahertz spectroscopy is shown in Table 1. The results, with consideration of errors, were basically similar. The errors mainly came from two aspects. Firstly, the terahertz spot center and the micrometer detection needle center were not be completely overlapped. Secondly, due to the existence of surface roughness in eroded areas, the contacting surface of the micrometer detection needle could just reach the height of the highest peak of the tested area, so the results obtained by the micrometer were larger than the actual value.

It was impossible to use the electronic digital readout micrometer for online monitoring, and the effect of roughness cannot be ignored. Nevertheless, for the first reflection peak, the abscissa changes corresponded to the thickness reduction, and the ordinate changes corresponded to the influence of the surface roughness. So, the terahertz method could overcome the drawbacks that the electronic digital readout micrometer had.

To characterize the interface morphology roundly, another three tested areas (D, E, F) were introduced. The distance between adjacent points was 2 mm, and all the points were kept on the same line, as shown in Figure 11.

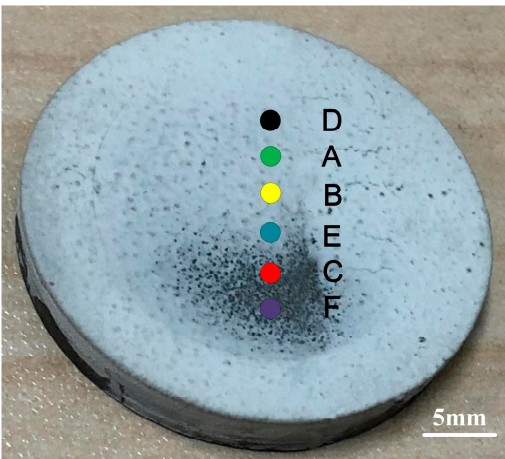

**Figure 11.** Schematic diagram of six tested areas' distribution.

As shown in Figure 12, the distances of adjacent points and loss of thickness, $\Delta d$, were indicated via the X and Z axis coordinates, respectively. The dotted red line of the loss thickness distribution characterized the basic profile of the tested areas.

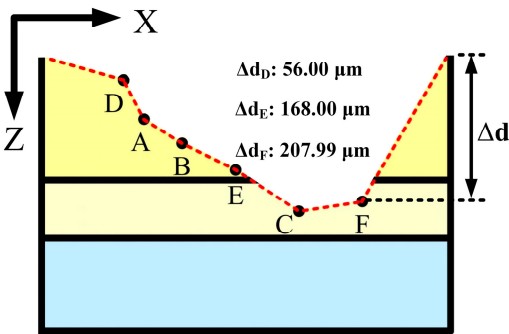

**Figure 12.** Schematic diagram of the interface morphology obtained by terahertz time-domain spectroscopy.

For the changes in the coordinate caused by roughness, the first reflections of eight tested areas (reference: Total specular reflection) were extracted, and the frequency characteristics, $F_S$, obtained by taking the FFT are shown in Figure 13.

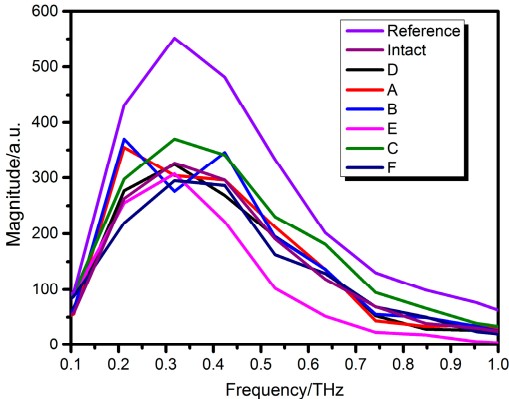

**Figure 13.** Frequency characteristics of reflected waves, S, obtained from eight tested areas.

Among these tested areas, the first reflection of the terahertz waves was reflected on the surface of the top coat in areas D, A, B, and E and was directly reflected on the surface of the bond coat in areas C, F.

The calculation of the roughness for the former (D, A, B, E) in Figure 12, followed Equation (8):

$$\left(\frac{F_S}{F_r \cdot \frac{n-1}{n+1}}\right)^2 = e^{-\left(\frac{4\pi\sigma f}{c}\right)^2} \tag{8}$$

The calculation of the roughness for the latter (C, F) in Figure 12, followed Equation (9):

$$\left(\frac{F_S}{F_r}\right)^2 = e^{-\left(\frac{4\pi\sigma f}{c}\right)^2} \tag{9}$$

$F_r$ is the reference frequency characteristics of the reflected waves S, and $n$ is the refractive index of the top coat without the influence of roughness ($n = 5.16$).

Combining with Figure 13, $R_z$ roughness of six tested areas were obtained by Equations (8) and (9) and all the $R_z$ roughness calculation results of the tested areas were greater than 29.30 μm, which meant that the eroded surface was no longer smooth again. $R_a$ roughness was the arithmetic mean of the absolute offset value and was chosen to correct the thickness loss. Figure 14 shows the thickness loss distribution with $R_a$ roughness.

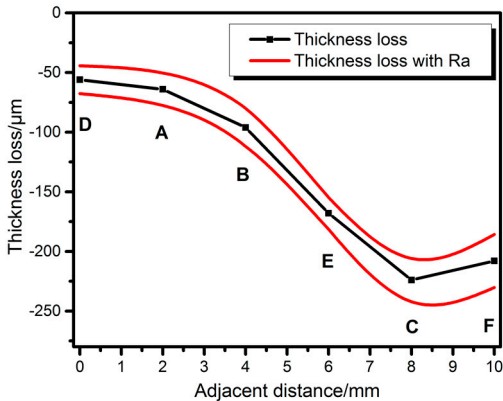

**Figure 14.** Thickness loss distribution with $R_a$ roughness.

By analyzing the time and frequency domain results of the first reflection peak, the information embodied in the top coat surface was obtained. Nevertheless, the characteristics of structural changes within the top coat after erosion were not yet clear. To observe internal cracks' and pores' changes, three regions, D, A, and B, were selected for analysis, because the top coat of the other three regions, E, C, and F, had basically been washed away.

When the brittle ceramic coating was impacted, the original cracks and pores inside the ceramic coating expanded and continued to sprout under external force, and the porosity of ceramic coating got increased [3,37,38]. In the presence of THz scattering, the pulse was broader with a more porous structure [39]. To investigate the relationship between the internal structure variation tendency and the broadening of the second reflection pulse, the ratio of the peak to peak value of the pulse was proposed to quantify the broadening of the pulses. In Table 2, the $\Delta t_{pp}$ ratio of the reference and different areas is presented. It was quite evident that the ceramic coating of different erosion areas had different $\Delta t_{pp}$ ratios. Compared to the reference signal, the pulses of different eroded areas were slightly broader regarding the first reflection. This was because the surface of different tested areas was not smooth, which causes slight scattering and resulted in pulse broadening. To eliminate the influence of the remaining thickness, the ratio was also divided by the remaining thickness obtained by the terahertz method. The comparison between the intact and eroded areas showed that the ratio of the stronger

pulses (2nd/reference)/2*d* broadened as the porosity of the ceramic coating became higher, which suggested that it was possible to use the ratio to inspect the inner structure evolution and porosity of the top coat.

**Table 2.** The ratio of $\Delta t_{pp}$ for intact and erosion samples having different porosities without any other treatment and divided by the thickness of the top coat.

| Tested Areas/$\Delta t_{pp}$ | 1st (ps) | 1st/Reference | 2nd/Reference | (2nd/Reference)/2*d* |
|---|---|---|---|---|
| Reference | 1.07 | 1 | – | – |
| Intact | 1.12 | 1.05 | 2.03 | 5.72 |
| D | 1.12 | 1.05 | 2.51 | 10.36 |
| A | 1.12 | 1.05 | 2.30 | 10.15 |
| B | 1.12 | 1.05 | 1.44 | 8.89 |

From the interface morphology characterized by the terahertz method, the increase or decrease of the erosion thickness in adjacent areas and maximum erosion thickness loss area (most dangerous area) could be determined efficiently. Moreover, terahertz time-domain spectroscopy technology offered early warning of erosion failure. It is also possible to simultaneously monitor the variation of the coating porosity, and further research is needed to fully understand the relation between the terahertz pulse broadening ratio and microstructures of the top coat.

## 4. Conclusions

In this work, the reflective THz-TDS system with an incident angle of 0 was used for the thickness measurement and interface morphology characterization of the YSZ topcoat before and after an erosion test, respectively. For the thickness measurement of the as-sprayed sample, the relative errors $(d_{THz} - d_{SEM})/d_{SEM}$ were 3.08% and 0.57% before and after considering the surface roughness of the top coat, respectively. The Kirchhoff equation and thickness measurement result showed that the surface of the as-sprayed coatings was smooth at the frequency of 0.32 THz. For the interface morphology characterization of the eroded sample, the interaction between THz waves and the TBCs was discussed with respect to the intact and erosion areas, and two different mathematical models were used to estimate the thickness before and after the erosion test. The thickness loss after the erosion test estimated by both terahertz waves and a micrometer were basically similar, but the latter ignored the surface roughness of TBCs. The surface roughness of eroded areas was greater than the critical value of 29.30 μm, which was estimated by the Kirchhoff equation. It meant that the eroded surface was not smooth again. Finally, not only was the basic interface morphology of the TBCs with surface roughness successfully characterized based on the THz time-domain and frequency-domain spectrum, but the evolution tendency of the internal structure and porosity of the top coat may potentially be evaluated by introducing the broadening ratio of the second reflection pulse. It is expected that the THz technique can be widely used to prevent and evaluate various forms of failure, and monitor the health of the TBCs thanks to its nondestructive, noncontact, and nonionizing features.

**Author Contributions:** Conceptualization, D.Y. and W.W.; Methodology, D.Y. and W.W.; Software, D.Y.; Validation, D.Y.; Formal Analysis, D.Y., W.W. and H.Z.; Investigation, D.Y., W.W., J.H., X.L. and H.Z.; Resources, W.W.; Data Curation, D.Y. and W.W.; Writing—Original Draft Preparation, D.Y. and W.W.; Writing—Review & Editing, D.Y. and W.W.; Visualization, D.Y.; Supervision, W.W.; Project Administration, W.W.; Funding Acquisition, W.W.

**Funding:** This research was funded by National Natural Science Foundation of China (No. 51775189) and Science and Technology Commission of Shanghai Municipality Project (No. 16DZ2260604).

**Acknowledgments:** The authors thank the support of the Shanghai key laboratory of modern optical systems, and University of Shanghai for Science and Technology.

**Conflicts of Interest:** The authors declare no conflict of interest.

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
