# Peer review of "Nondestructive Interface Morphology Characterization of Thermal Barrier Coatings Using Terahertz Time-Domain Spectroscopy"

_coatings, doi:10.3390/coatings9020089_

Reviewer 1 Report

The role of TGO layer on terahertz wave reflections/interactions is ignored.

The erosion test using steam and particles does not specify temperature and flow velocity conditions. These conditions should be engine relevant. A hot burner test rig with sand particle ingestion should have been used to be realistic. TGO growth is dependent on the duration of exposure to high temperature flow over coated samples. The paper lacks testing details. It is difficult to judge if the erosion test performed, simulated, in fact, realistic hot particulate flow conditions.

There is lack of information on the laser spot size. How much interference is present if the erosion measurement spots on the same sample are too close? How much should be the minimum separation between spots?

Why not use Confocal Laser Microscopy to compare results from the THz time-domain spectroscopy method instead of a micrometer?

Spot measurements on the sample are cumbersome to do to get the complete eroded surface profile for real turbine blade shapes. There is no mention of curved surface effects on THz wave reflections and interactions.

Author Response

Response to the reviewer #1

Thank you very much for the valuable comments on our paper entitled “Nondestructive Interface Morphology Characterization of Thermal Barrier Coatings using Terahertz time-domain spectroscopy”. Following are the explanations concerned to your comments.

Point 1: The role of TGO layer on terahertz wave reflections/interactions is ignored. 

Answer: The role of TGO layer is vital to be considered on non-destructive evaluation of TBCs, but it doesn't not contain TGO layer in as-sprayed coatings. It doesn't mean that terahertz method cannot be used to inspect the TGO layer and some scholars had done some work on this part previously (Ref 26-30). Even if there is TGO layer inside the eroded coatings, according to the study from these scholars, the thickness of TGO layer can also be estimated and the location of first terahertz wave reflection will not be changed, so the thickness loss which obtained by first reflection will also not be changed. Similarly, the broadening ratio of the second terahertz wave reflection which used to evaluate the internal structure variation of TC will still available, because the TGO layer just contributes to the time delay change of the second terahertz wave reflection and will not change the broadening ratio.

Point 2: The erosion test using steam and particles does not specify temperature and flow velocity conditions. These conditions should be engine relevant. A hot burner test rig with sand particle ingestion should have been used to be realistic. TGO growth is dependent on the duration of exposure to high temperature flow over coated samples. The paper lacks testing details. It is difficult to judge if the erosion test performed, simulated, in fact, realistic hot particulate flow conditions.

Answer: This erosion test is only for obtaining the sample after erosion and the erosion test is carried out at room temperature. The testing details has been added, including the schematic diagram of the erosion device, the nozzle-to-substrate distance and so on.

Section 2.3:“……The schematic is shown in Figure 3…….and a 6 cm nozzle-to-substrate distance was used. To obtain a well-defined geometry erosion scar, 20 mm diameter opening was used to mask the specimen surface……”

Point 3: There is lack of information on the laser spot size. How much interference is present if the erosion measurement spots on the same sample are too close? How much should be the minimum separation between spots?

Answer: The spot diameter is 2 mm and the center spacing between adjacent test areas is also 2 mm. In addition, the measurement on the eroded coating is done one by one. Correspondingly the interference does not exist. These information has been added.

Section 2.1: “……The spot diameter of terahertz waves is 2 mm…...”

Point 4: Why not use Confocal Laser Microscopy to compare results from the THz time-domain spectroscopy method instead of a micrometer?

Answer: As mentioned above, the spot diameter of terahertz waves is 2mm. In order to ensure that the monitoring areas of different test methods are consistent, a micrometer with 2mm diameter test needle is used to estimate the thickness loss. So that the result obtained by micrometer make comparison with the results obtained by terahertz method in this paper.

Point 5: Spot measurements on the sample are cumbersome to do to get the complete eroded surface profile for real turbine blade shapes. There is no mention of curved surface effects on THz wave reflections and interactions.

Answer: With the development of terahertz technology, smaller spot and faster scanning speed terahertz inspection equipment will come out or may have come out, so I think that problem will be solved in the near future. Future research trials will seek more advanced equipment to complete.

The sample in this paper is flat, but the real turbine blade curved surface is not flat in reality. From the beginning of the initial intact thickness observation, there is a trouble problem. The trouble problem is that when we want to estimate the thickness of TC before erosion test using terahertz waves with an angle of incidence of 0, in fact, the real angle of incidence is not 0 in most cases. This problem need to be solved. Now our idea is to get the spot distance between the 1st reflection and 2nd reflection. If the real angle of incidence is 0, the spot distance is also 0. So the spot distance can be used to estimate the actual incident angle. Similarly, this problem also need the improved terahertz inspection equipment. And it will be the future research interest for us, thank you for your good suggestions.

Section 3.2: “……Nevertheless, the ceramic coatings was deposited on flat surface here, but the real turbine blade curved shape can’t keep the incident angle at 0 in most cases. This problem needs to be solved and the future research will focus on it.”

Reviewer 2 Report

The authors of the paper: “Nondestructive Interface Morphology 3 Characterization of Thermal Barrier Coatings using 4 Terahertz time-domain spectroscopy” proposing a THz method for characterization of thermal coatings. I agree with them that the THz method can be very valuable method for the community. However, the key experiment is very badly described, there is plenty of information missing on the description of the experimental setup (individual components) as well as very important size of the THz spot on the surface. This are important information for proper analysis of their data. Furthermore, in the photographs the scale bars are missing. For these reasons I would like to see this information before accepting the paper in the journal.

Author Response

Response to the reviewer #2

Thank you very much for the valuable comments on our paper entitled “Nondestructive Interface Morphology Characterization of Thermal Barrier Coatings using Terahertz time-domain spectroscopy”. Following are the explanations concerned to your comments.

Point 1: “Nondestructive Interface Morphology 3 Characterization of Thermal Barrier Coatings using 4 Terahertz time-domain spectroscopy” proposing a THz method for characterization of thermal coatings. I agree with them that the THz method can be very valuable method for the community. However, the key experiment is very badly described, there is plenty of information missing on the description of the experimental setup (individual components) as well as very important size of the THz spot on the surface. This are important information for proper analysis of their data. Furthermore, in the photographs the scale bars are missing. For these reasons I would like to see this information before accepting the paper in the journal.

Answer: We have made supplement on the details of erosion test in this paper, including the schematic diagram of the erosion device, the nozzle-to-substrate distance and so on. The THz spot diameter is 2mm and the center spacing between adjacent test areas is also 2mm. In order to ensure that the monitoring areas of different test methods are consistent, a micrometer with 2mm diameter test needle is used to estimate the thickness loss. The quality of the photographs was improved. These information has been added to the manuscript.

Section 2.1: “The spot diameter of terahertz waves is 2 mm. Each data is obtained by 256 scans averaging to ensure the reliability……The test temperature is 20 oC……”

Section 2.3: “……The schematic is shown in Figure 3……and a 6 cm nozzle-to-substrate distance was used. To obtain a well-defined geometry erosion scar, 20 mm diameter opening was used to mask the specimen surface……”

Round  2

Reviewer 1 Report

Thank you for answering all the queries and making revisions to your paper. Hope that this research will be further developed for real turbine blades with curved shape in the near future. I agree that this a good first step that can lead to a better NDE method for determining CMAS deposition in turbine blades.

Reviewer 2 Report

I am fine with improved version of the manuscript.

Author Response

Response to the Reviewer

Thank you very much for the valuable comments on our paper entitled “Nondestructive Interface Morphology Characterization of Thermal Barrier Coatings using Terahertz time-domain spectroscopy”.